# WearVox: An Egocentric Multichannel Voice Assistant Benchmark for Wearables

**Zhaojiang Lin**[1*], **Yong Xu**[1*], **Kai Sun**[1*], **Jing Zheng**[1], **Yin Huang**[1], **Surya Teja Appini**[1], **Krish Narang**[1], **Renjie Tao**[1], **Ishan Kapil Jain**[1], **Siddhant Arora**[1,2†], **Ruizhi Li**[1], **Yiteng Huang**[1], **Kaushik Patnaik**[1], **Wenfang Xu**[1], **Suwon Shon**[1], **Yue Liu**[1], **Ahmed A Aly**[1], **Anuj Kumar**[1], **Florian Metze**[1], **Xin Luna Dong**[1]

[1]Meta,    [2]Carnegie Mellon University
zhaojiang@meta.com, sunkaicn@meta.com, yongxu@meta.com,
lunadong@meta.com

## Abstract

Wearable devices such as AI glasses are transforming voice assistants into always-available, hands-free collaborators that integrate seamlessly with daily life, but they also introduce challenges like egocentric audio affected by motion and noise, rapid micro-interactions, and the need to distinguish device-directed speech from background conversations. Existing benchmarks largely overlook these complexities, focusing instead on clean or generic conversational audio. To bridge this gap, we present WearVox, the first benchmark designed to rigorously evaluate voice assistants in realistic wearable scenarios. WearVox comprises 3,842 multi-channel, egocentric audio recordings collected via AI glasses across five diverse tasks including Search-Grounded QA, Closed-Book QA, Side-Talk Rejection, Tool Calling, and Speech Translation, spanning a wide range of indoor and outdoor environments and acoustic conditions. Each recording is accompanied by rich metadata, enabling nuanced analysis of model performance under real-world constraints. We benchmark leading proprietary and open-source speech Large Language Models (SLLMs) and find that most real-time SLLMs achieve accuracies on WearVox ranging from 29% to 59%, with substantial performance degradation on noisy outdoor audio, underscoring the difficulty and realism of the benchmark. Additionally, we conduct a case study with two new SLLMs that perform inference with single-channel and multi-channel audio, demonstrating that multi-channel audio inputs significantly enhance model robustness to environmental noise and improve discrimination between device-directed and background speech. Our results highlight the critical importance of spatial audio cues for context-aware voice assistants and establish WearVox as a comprehensive testbed for advancing wearable voice AI research.[1]

## 1 Introduction

Wearable devices, such as AI glasses, are transforming voice assistants from handheld tools into always-available, body-worn collaborators. Unlike phones and smart speakers where interactions are episodic, hands-free only by choice, and typically occur in acoustically stable environments, wearables operate at the edge of our attention, seamlessly integrating with daily activities like walking, commuting, and socializing. While this convenience unlocks new possibilities, it also introduces unique challenges: egocentric audio affected by motion and wind noise, rapid micro-interactions constrained by strict latency requirements, and the need to distinguish device-directed requests from side speech and background noise. Yet, existing benchmarks such as VoiceBench (Chen et al., 2024), Spoken-CoQA (You et al., 2022), and Spoken-SQuAD (Lee et al., 2018) focus primarily on clean audio or generic conversational scenarios, overlooking the specific complexities inherent to wearable interactions.

---

[*]Joint first author
[†]Work done at Meta
[1]WearVox is available at https://github.com/facebookresearch/wearvox.

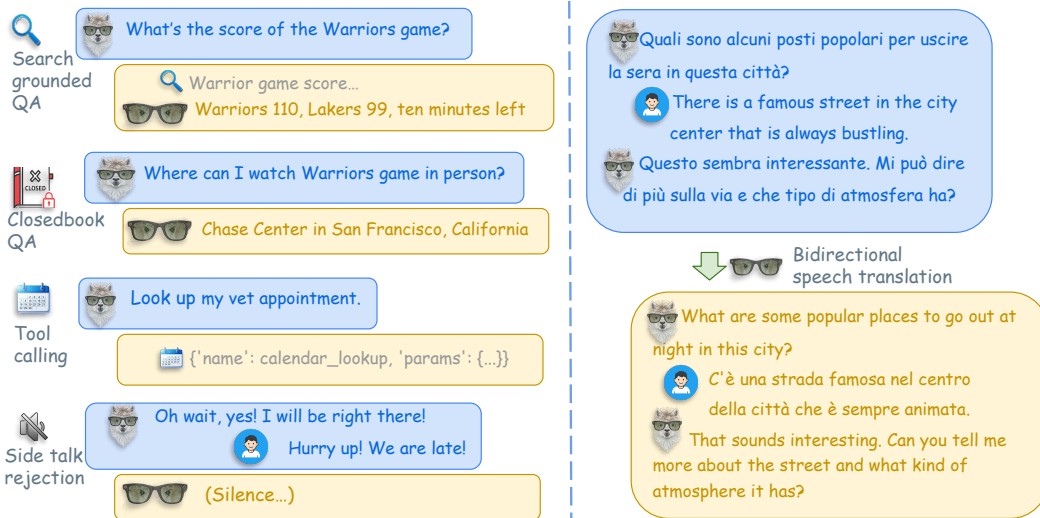

Figure 1: Examples of tasks from the WearVox dataset. The audio queries are recorded with AI glasses (transcribed in blue). The ground truth for each task is provided in text format.

To bridge this critical gap, we introduce 𝕎𝕖𝕒𝕣𝕍𝕠𝕩, the first wearable-specific voice assistant benchmark designed to rigorously evaluate state-of-the-art speech Large Language Models (SLLMs) in realistic wearable scenarios. WearVox comprises a comprehensive collection of 3,842 multi-channel, egocentric audio recordings, spanning 5 different tasks that reflect practical situations encountered by users of wearable devices such as AI glasses, both indoors and outdoors. WearVox is distinguished by several valuable features:

1. **Ego-centric, multi-channel audio**: All recordings in WearVox are captured from a first-person perspective using AI glasses equipped with multiple microphones, simulating the audio input typical of wearable devices. The dataset is designed to reflect the complexity of real-world interactions, featuring a variety of speaker roles including the primary glasses wearer, conversational partners, and bystanders who positioned at different angles and distances. This setup enables the modeling of realistic conversational dynamics, such as direct queries, interruptions, side-talk, and non-assistant-directed speech, which are essential for robust and context-aware voice assistant performance.

2. **Comprehensive environmental and acoustic coverage**: WearVox covers a wide range of indoor and outdoor environments, including office spaces, cafés, cars, as well as streets, parks, and construction zones. Approximately 31% of the dialogues were recorded indoors, while 63% took place outdoors. Recordings were conducted under both quiet and noisy conditions, with 58% of the data collected in noisy environments and 42% in quiet settings. The dataset features 13 different noise types, such as rustling leaves and construction noise, carefully selected to represent real-world scenarios. Each audio sample is accompanied by detailed metadata describing participant positions, distances, and environmental context, ensuring that the dataset captures the nuanced challenges inherent to wearable audio.

3. **Diverse and realistic wearable assistant tasks**: The benchmark encompasses a broad array of wearable assistant tasks, including Search-Grounded Question Answering (QA), Closed-Book QA, Side-Talk Rejection, Tool Calling, and Speech Translation. The dataset is meticulously curated to reflect the functionalities expected of next-generation wearable assistants, ensuring that models are evaluated across a wide spectrum of practical and challenging scenarios.

Building on the WearVox benchmark, we conduct comprehensive experiments to evaluate the performance of state-of-the-art open-source and proprietary SLLMs in realistic wearable scenarios. Our evaluation includes leading models such as GPT-4o (Hurst et al., 2024) and Gemini 2.5-flash (Comanici et al., 2025), as well as open-source models like Qwen-2.5 omni (Xu et al., 2025), Gemma 3n (Team, 2025a), Phi-4 multimodal (Abouelenin et al., 2025), and Kimi-Audio (Ding et al., 2025).

For a robust comparison, we also include a two-stage pipeline combining Whisper ASR (Radford et al., 2023) with a text LLMs GPT-5 (OpenAI, 2025). This diverse set of models allows us to systematically analyze the strengths and limitations of current SLLMs when faced with the unique challenges posed by egocentric, multi-channel wearable audio. The experimental results show that most real-time SLLMs achieve accuracies on WearVox ranging from 29% to 59%, with substantial performance degradation on noisy outdoor audio, underscoring the difficulty and realism of the benchmark.

To assess the impact of multi-channel audio signal in wearable scenarios, we conduct a case study with two new SLLMs: one utilizing single-channel audio and another leveraging a multi-channel approach built on the Llama 4 Scout (Team, 2025b) architecture. Our findings reveal that incorporating multi-channel audio inputs greatly improves model resilience to environmental noise and enhances the ability to differentiate between device-directed speech, side conversations, and background noise. Specifically, side talk rejection accuracy increased from 85.6% to 93.9%, while overall accuracy improved from 61.9% to 66.4%. These results highlight the critical importance of spatial audio cues for enabling context-aware voice assistants in wearable applications.

## 2 RELATED WORK

**Voice Assistant Benchmarks.** The trajectory of voice assistant benchmarking has evolved from ASR-dependent comprehension tasks to comprehensive, end-to-end evaluation of speech LLMs. Early datasets such as Spoken-SQuAD (Lee et al., 2018) and Spoken-CoQA (You et al., 2022) extended SQuAD and CoQA into the spoken domain by generating speech with TTS. Subsequent benchmarks, including HeySQuAD (Wu et al., 2023) and LibriSQA (Zhao et al., 2024), improved realism by incorporating large-scale human-spoken recordings, reducing reliance on synthetic speech. More recent efforts—AudioBench (Wang et al., 2025), MMAU (Sakshi et al., 2024), VoiceBench (Chen et al., 2024), and CAVA (Held et al., 2025)—broaden the scope beyond QA to include speech instruction following, paralinguistic understanding, acoustic scene perception, and even music reasoning, reflecting the expanding capabilities of Audio-LLMs. Finally, the latest benchmarks such as CAVA (Held et al., 2025) and FDX-bench (Lin et al., 2025) emphasize real-time conversational aspects, explicitly evaluating turn-taking, latency, and duplex interaction—crucial properties for practical voice assistants. In contrast, WearVox is the first voice benchmark designed specifically for wearable computing, leveraging egocentric multi-channel audio and a diverse range of real-world environments to capture conversational dynamics (such as side talk and non-assistant-directed speech) that are critical for advancing next-generation voice assistants. A more detailed comparison between WearVox and existing voice assistant benchmarks is provided in Table 1.

**Speech LLMs.** Modern speech LLMs push beyond ASR-LLM-TTS pipelines toward end-to-end, streaming "omni" assistants. GPT-4o (Hurst et al., 2024) integrates audio understanding and generation with real-time performance, catalyzing speech-first UX in production systems. Gemini 2.5 Flash (Comanici et al., 2025) adds native audio support and controllable "thinking" budgets, balancing latency and reasoning for live dialog. In the open-weights space, Qwen2.5-Omni (Xu et al., 2025), Kimi-Audio (Ding et al., 2025), GLM4-Voice (Zeng et al., 2024) and Phi-4 Multimodal (Abouelenin et al., 2025) provide unified speech-text modeling with competitive audio-reasoning performance. Gemma 3n (Team, 2025a) extends this trend to mobile-scale audio understanding. Full-duplex LLMs such as Moshi (Défossez et al., 2024), SALMONN-omni (Yu et al., 2024), and SyncLLM (Veluri et al., 2024) enable listen and speak at the same time without strict turn-taking. As a complement, we introduce a multichannel SLLM and share findings on the role of spatial audio, offering insights beyond the single-channel focus of existing speech LLMs.

## 3 WearVox DATASET

### 3.1 PROBLEM DEFINITION

The WearVox dataset is developed to benchmark and advance the capabilities of wearable voice assistants in real-world, egocentric audio environments. It provides a comprehensive suite of tasks (as illustrated in Figure 1), diverse speaker roles, and varied acoustic conditions to facilitate robust evaluation and development of next-generation voice assistant systems.

| Benchmark | Egocentric Audio | Multi-channel Audio | Conversational Dynamics | Domain Diversity | Dataset Size | Audio Source |
|---|---|---|---|---|---|---|
| Spoken-SQuAD | × | × | × | × | 42K | TTS |
| Spoken-CoQA | × | × | × | ✓ | 40K | TTS |
| HeySQuAD | × | × | × | × | 173K | TTS/recording |
| LibriSQA | × | × | × | ✓ | 214K | LibriSpeech |
| AudioBench | × | × | × | ✓ | 5.5K | LibriSpeech/Clotho |
| MMAU | × | × | × | ✓ | 10K | Diverse public audio |
| VoiceBench | × | × | × | ✓ | 5.8K | TTS/recording |
| CAVA | × | × | partial | ✓ | 6K | STOP |
| FDX-Bench | × | × | ✓ | × | <1K | Candor/TTS |
| WearVox | ✓ | ✓ | ✓ | ✓ | 3.8K | Consumer wearables |

Table 1: Comparison of WearVox to existing voice assistant benchmarks.

### 3.1.1 TASKS FORMULATION

WearVox encompasses five core tasks that reflect both common and challenging scenarios for wearable voice assistants. All tasks are formulated as a **Text In, Speech In, Text Out** problem, defined as:

$$f(T_I, S_I) \rightarrow T_O$$

where $T_I$ is the input text prompt, $S_I$ is the input speech signal, and $T_O$ is the output text. Each task has a distinct fine-grained composition, as detailed below:

1. **Search-Grounded QA.** Many daily queries to wearable assistants (e.g., financial news, sports scores) require up-to-date, external information. In this task, the assistant must provide factual answers based on search results.
   - $T_I$: Task description and external search results
   - $S_I$: Wearer request in speech
   - $T_O$: Answer in text

2. **Closed-Book QA.** The assistant responds to general knowledge questions without access to external resources, relying solely on its internal knowledge.
   - $T_I$: Task description
   - $S_I$: Wearer request in speech
   - $T_O$: Answer in text

3. **Tool Calling.** The assistant is required to invoke specific tools or APIs (e.g., music player, reminders) based on wearer requests.
   - $T_I$: Task description and tool/function definitions (including tool name, tool description, and parameters)
   - $S_I$: Wearer request in speech
   - $T_O$: Tool call in JSON format

4. **Side Talk Rejection.** The system must accurately distinguish and ignore non-device-directed speech, such as background conversations and bystander chatter.
   - $T_I$: Task description
   - $S_I$: Side talk speech, triggered by either the wearer, conversational partner, or bystanders
   - $T_O$: Special control token (e.g., [Mute]) to suppress downstream components (such as TTS)

5. **Bidirectional Speech Translation.** The assistant facilitates translation between the wearer and a conversational partner who speak different languages. In this task, the assistant must perform both speaker diarization and speech translation simultaneously. We focus on offline, whole-dialog translation rather than simultaneous translation, simplifying the evaluation protocol under the assumption that performance in both settings is highly correlated.
   - $T_I$: Task description
   - $S_I$: Bilingual, multi-turn dialogue between the wearer and conversational partner
   - $T_O$: Diarized, translated dialogue in text

### 3.1.2 SPEAKER ROLES

WearVox simulates realistic, multi-party interactions by involving three distinct speaker roles:

- **Wearer**: The primary user of the wearable device, who initiates most device-directed queries and commands. In Search-Grounded QA, Closed-Book QA, and Tool Calling tasks, the wearer is typically the source of the spoken input ($S_I$), issuing questions or requests to the assistant. In Side Talk Rejection, the wearer may also produce non-device-directed speech, testing the assistant's ability to distinguish between intentional and incidental input. For Bidirectional Speech Translation, the wearer participates as one of the two parties in the bilingual conversation, requiring the assistant to correctly identify and translate their utterances.

- **Conversational Partner**: An individual actively engaged in dialogue with the wearer. This role is especially prominent in the Bidirectional Speech Translation task, where the conversational partner speaks a different language and participates in multi-turn exchanges with the wearer. The assistant must perform speaker diarization to attribute each utterance correctly and provide accurate translations for both parties.

- **Bystander**: A third-party speaker who may contribute incidental or background speech, simulating real-world distractions. The bystander's role is most critical in the Side Talk Rejection task, where their speech serves as a test for the assistant's ability to filter out non-device-directed input. Bystanders may also be present in other tasks, adding complexity to the audio environment and challenging the assistant's speaker identification and intent recognition capabilities.

### 3.1.3 ACOUSTIC CONDITIONS

The dataset includes recordings from a wide variety of environments, to capture the full spectrum of acoustic conditions encountered by wearable voice assistants.

**Indoor Environments:**  Recordings are conducted in rooms of varying sizes (small, medium, and large), offices, and busy hallways. These settings introduce a range of reverberation levels, background conversations, and ambient noises such as air conditioning or office equipment. Such conditions are particularly relevant for tasks like Search-Grounded QA and Closed-Book QA, where the assistant must accurately process user queries despite potential acoustic interference.

**Outdoor and Mobile Scenarios:**  Sessions take place in parks, picnic areas, parking lots, cars, and near construction zones. These environments introduce dynamic background noises, including wind, traffic, and construction sounds, which can mask or distort wearer speech.

**Noise Diversity and Signal-to-Noise Ratios:**  The dataset systematically varies the signal-to-noise ratio (SNR) by including both quiet scenarios (e.g., soft whispers, rustling leaves) and high-noise situations (e.g., vacuum cleaners, subways, buses, motorcycles). This diversity ensures that the assistant's performance can be evaluated across a continuum from controlled, low-noise environments to highly challenging, real-world auditory scenes.

### 3.2 DATA COLLECTION

The process comprises three key stages: script collection, egocentric audio recording, and ground truth annotation. Each stage is carefully structured to maximize the realism, utility and reliability of the resulting dataset.

**Script Collection**   The central objective in script collection is to ensure that the dataset authentically represents real-world use cases. For spoken QA, we curate questions from the CRAG (Yang et al., 2024) and Head-to-tail (Sun et al., 2024) datasets, categorizing them into popular, static factual questions for closed-book QA, and long-tail, rapidly changing factual questions for search-grounded QA. For the remaining three tasks, we first design several representative scenarios for each, then employ annotators to expand and construct multi-turn conversations based on these scenarios. For example, in the bidirectional speech translation task, seed scenarios typically involve a

foreigner approaching a local to ask questions on topics such as finding locations, accommodations, transportation, and reservations. In the tool calling task, we provide 8 predefined tools including calendar, web search, local search, music player etc. Annotators, with the assistance of LLMs (e.g., Llama 3.3 70B), create multi-turn conversations based on these scenarios and domains.

**Egocentric Audio Recording**  With the scripts prepared, the next step involves capturing egocentric multichannel audio data from glasses. To this end, we recruit a diverse group of native speakers: for the speech translation task, we hire native speakers of Italian, Spanish, Portuguese, German, and French who also understand English scripts; for the other tasks, we engage native English speakers. For each session, 2–3 individuals collaborate to simulate realistic interactions based on the provided scripts. Importantly, scripts serve as references during audio recording to enhance data quality; speakers are encouraged to follow the script loosely to ensure that the recorded speech sounds natural and conversational, rather than read verbatim. Details are available in Appendix A.3

**Ground Truth Annotation**  After data collection, we instructed our annotators to generate ground truth annotations for each dialogue. For the speech translation task, annotators transcribe the audio and provide corresponding translations based on the scripts and recordings. For the tool invocation task, annotators specify the appropriate API calls for each interaction. For spoken QA, we primarily reuse labels from the original CRAG and Head-to-Tail datasets. For non-device-directed speech samples, we assign a special [Mute] token to indicate that these queries should be ignored as invalid.

## 3.3 DATASET STATISTICS

We collected 3,842 dialogues with egocentric multichannel audio recordings, comprising 547 Search-Grounded QA, 588 Closed-Book QA, 1,082 Side-Talk [2], 1,125 Tool Calling, and 1,000 Translation tasks. Approximately [3] 31% of the recordings took place indoors, while 63% were recorded outdoors. In terms of noise conditions, 58% of the recordings were made in noisy environments, and 42% in quiet settings. A more detailed breakdown of environment and noise type distributions is provided in Appendix A.4.

## 4 BENCHMARKING

In this section, we systematically evaluate state-of-the-art SLLMs on the WearVox benchmark to assess their capabilities and limitations in addressing the unique challenges of wearable contexts.

### 4.1 EXPERIMENTAL SETUP

#### 4.1.1 BASELINES

We consider both proprietary and open-source models in our evaluation.

- **Open-Source Models:** Gemma 3n (Team, 2025a), Kimi-Audio (Ding et al., 2025), Qwen2.5-Omni (Xu et al., 2025)
- **Proprietary Models:** GPT-4o Audio [4] (Hurst et al., 2024), Gemini 2.5 Flash (Comanici et al., 2025), GPT-5 w/ Whisper (OpenAI, 2025)

Since the existing state-of-the-art SLLMs are trained on single-channel audio, we follow previous work (Lin et al., 2024; Xie et al., 2025) applying beamforming to convert the multichannel recordings into a single channel for evaluating single-channel SLLM performance.

#### 4.1.2 EVALUATION SETTINGS

To facilitate comprehensive and consistent evaluation across diverse tasks, we divide our assessment into two settings: turn-based and session-based evaluation.

---

[2]The Side Talk Rejection task contains 582 side talk and 500 valid queries duplicated from tool-calling tasks.

[3]Speech translation samples are excluded from this statistic due to missing audio metadata.

[4]https://platform.openai.com/docs/models/gpt-4o-audio-preview

| Baselines | Search Grounded QA | Closedbook QA | Tool Calling | Side Talk Rejection | Turn-based Micro-avg | Speech Translation |
|---|---|---|---|---|---|---|
| Gemma 3n | 29.4 | 20.4 | 5.7 | 59.9 | 29.7 | 14.8* |
| Kimi-Audio | 10.1 | 31.5 | 63 | 47.0 | 43.6 | 41.8* |
| Qwen2.5-Omni | 35.8 | 29.8 | 7.3 | 60.4 | 33.1 | 43.9* |
| GPT-4o Audio | 50.5 | 59.4 | 8.9 | 66.0 | 43.1 | 76.0 |
| GPT-5 w/ Whisper | 57.8 | 70.6 | 35.7 | 73.8 | 57.8 | 92.9* |
| Gemini 2.5 Flash | 49.0 | 46.8 | 44.4 | 88.2 | 59.8 | 50.3 |
| Gemini 2.5 Flash Thinking | 48.8 | 61.4 | 68.1 | 91.4 | 71.3 | 70.1 |

Table 2: Benchmarking results for both open-source and proprietary SLLMs on four turn-based tasks (including the micro-average across all turn-based tasks) and a session-based speech translation task. *Note: In the speech translation task, input audio for Gemma 3n, Kimi-Audio, and Qwen2.5 Omni is truncated at 30 seconds due to audio encoder context limits. In contrast, the GPT-5 w/ Whisper baseline transcribes dialogues turn-by-turn using ground truth segmentation and diarization labels.

**Turn-based Evaluation** Turn-based evaluation is applied to tasks such as Search Grounded QA, Closed-book QA, Tool Calling, and Side Talk Rejection, where model answer accuracy is assessed at each turn. For Search Grounded QA and Closed-book QA, we employ an LLM-based judge that references annotated ground truth responses to evaluate answer quality. In the Tool Calling task, we utilize Abstract Syntax Tree (AST) evaluation, following the methodology described in Patil et al. (2024), to rigorously compare the structure and content of predicted tool calls. For Side Talk Rejection, performance is measured using binary accuracy, indicating whether the model correctly identifies and suppresses non-device-directed speech. Tool Calling and Side Talk task share the same task prompt as shown in Appendix A.1. Thus, the model must generate a tool call for valid requests and produce a special control token to handle side talk.

**Session-based Evaluation** Session-based evaluation is used for the speech translation task to assess translation quality over entire dialog sessions. We provide ground truth translations and prompt an LLM judge to score each turn based on the quality of speaker diarization and translation. The final session score is computed by averaging the turn-level scores, with penalties applied for missing or hallucinated turns. The LLM judge prompts and the session-level score aggregation function are detailed in Appendix A.2.

**LLM Judge Quality Validation** The LLM-based judge for QA tasks was adapted from the auto-evaluation of CRAG, which has demonstrated over 98% agreement with human evaluation. For the translation task, we validated the judge on 200 randomly sampled examples and observed a strong Pearson correlation ($r = 0.89$) between our judge's scores and human ratings, with human raters using the same scoring scale as the judge.

## 4.2 MAIN RESULTS

Table 2 reports the main results of WearVox, highlighting the substantial variability in performance across tasks and models. Open-source baselines, including Gemma 3n, Kimi-Audio, and Qwen2.5-Omni, generally underperform, particularly in search grounded QA and tool calling. This underperformance can be partially attributed to their relatively smaller model sizes (fewer than 8B parameters), which limits their reasoning capabilities across different modalities, such as user audio and text context. In contrast, proprietary SLLMs demonstrate more balanced performance. Both GPT-4o Audio and Gemini 2.5 Flash achieve overall scores above 40, with GPT-4o Audio excelling in speech translation (76.0%) and Gemini 2.5 Flash exhibiting strong robustness to side-talk (88.2%). However, we observe that GPT-4o Audio occasionally ignores the task prompt and generates direct responses in the tool calling task, resulting in low tool calling accuracy (8.9%). We hypothesize that GPT-4o Audio is specifically trained to handle audio input and output, and that structured text output capability is not fully optimized. GPT-5 with Whisper achieves the best search grouned QA (57.8%) and closed-book QA (70.6%), yielding a strong overall turn-based accuracy of 57.8%.

Enabling thinking mode in Gemini 2.5 Flash significantly improves performance on four out of five tasks, increasing overall turn-based task accuracy from 59.8% to 71.3% and speech translation

| Task | Gemini 2.5 Flash | Gemini 2.5 Flash Thinking | GPT-4o Audio |
|---|---|---|---|
| Closedbook QA | 1368.69 | 2287.76 | 1220.22 |
| Search Grounded QA | 1526.56 | 9194.94 | 1867.66 |
| Speech Translation | 2138.11 | 11321.49 | 7523.24 |
| Side Talk Rejection | 1306.62 | 2176.97 | 1341.04 |
| Tool Calling | 1404.69 | 2084.19 | 1289.99 |

Table 3: Time to First Token (TTFT) breakdown per task for Gemini 2.5 Flash (with and without thinking mode) and GPT-4o Audio. All values are reported in milliseconds (ms).

accuracy from 50.3% to 70.1%. However, it is important to note that thinking mode introduces substantial latency, as extensive reasoning tokens are generated prior to producing the actual response.

In our experiments, we observed a substantial increase in time to first (response) token (TTFT) latency for Gemini 2.5 Flash in Thinking mode, averaging 5546 ms compared to 1592 ms in non-Thinking mode. In Table 3 we report the per-task TTFT breakdown for Gemini 2.5 Flash with and without thinking mode, as well as GPT-4o Audio. We observe that the thinking model exhibits significantly higher latency compared to its non-thinking counterpart, primarily due to the overhead of thinking token generation. GPT-4o Audio demonstrates comparable latency to Gemini 2.5 Flash across most tasks, with the notable exception of speech translation, where the slower audio encoding during prefill likely contributes to increased delay. Overall, the latency difference between thinking and non-thinking model could significantly impact the user experience on wearable devices. Balancing the trade-off between real-time responsiveness and response quality remains an important direction for future research.

### 4.3 CASE STUDY: MULTICHANNEL SLLMS

Beyond existing single-channel SLLMs, we present a case study in which we develop a multichannel SLLM and compare its performance to its single-channel counterpart. Our primary research question is: **Does multichannel audio provide additional value over the beamformed audio channel in real-world wearable voice assistant tasks?**

We construct our SLLM by building on Llama-4-Scout-17B-16E (Team, 2025b) and a 1B parameter Conformer (Gulati et al., 2020) speech encoder that pre-trained with BEST-RQ (Chiu et al., 2022) as in Llama3 speech (Dubey et al., 2024). For training, we follow the speech alignment methodology described in AudioChatLlama (Fathullah et al., 2024). We begin with automatic speech recognition (ASR) data and prompt Llama-4-Scout-17B-16E to generate responses based on the corresponding text transcripts. The original audio is then paired with these generated responses to create synthetic speech QA data. Both the speech QA and ASR datasets are used to train LLM, along with an audio feature projection layer, while keeping the speech encoder frozen. More implementation details are available in Appendix A.6

To enable native multichannel audio processing, we convert all the original single-channel audio into simulated five-channel recordings, based on the microphone array configuration of AI glasses (Meta, 2024). Room impulse responses (RIRs) from real environments are used to model spatial diversity. We further augment the data by adding indoor noise from a diverse corpus at random signal-to-noise ratios (SNRs) ranging from -5 dB to 40 dB. Additionally, we introduce varying overlap ratios of bystander speech into the multichannel mixtures to simulate realistic acoustic conditions. We use the beamformed single-channel audio to train a single-channel SLLM, which we refer to as **SC WearLlama** (Single Channel Wearable Llama). In contrast, the model trained on multi-channel audio is denoted as **MC WearLlama**. As illustrated in Figure 2, unlike the SC WearLlama, which processes only the beamformed audio channel (c_x), the MC WearLlama processes both channel 0 (c_0), typically the channel with the highest SNR, and the beamformed channel in an interleaved manner.

Table 4 compares SC WearLlama and MC WearLlama on the WearVox benchmark. MC WearLlama shows a clear improvement in tool calling (63.9% vs. 58.5%) and side-talk rejection (93.9% vs. 85.4%), indicating that spatial audio cues help the model better separate user-directed speech from background interference and execute device-control tasks more reliably. However, both models

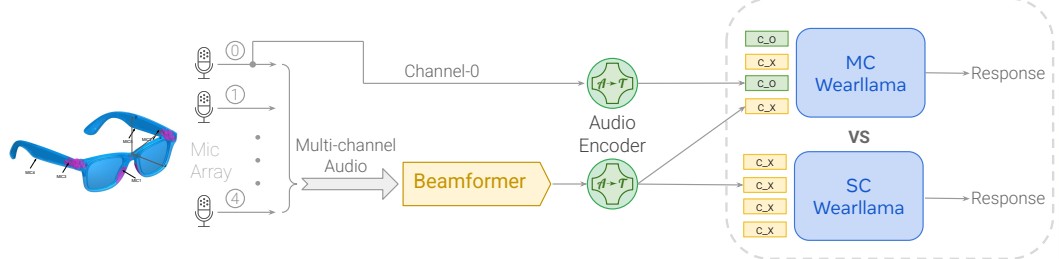

Figure 2: Illustration of SC WearLlama and MC WearLlama inference. SC WearLlama encodes only the beamformed audio channel (c_x), whereas MC WearLlama processes both channel 0 (c_0),typically the channel with the highest SNR, and the beamformed channel in an interleaved manner.

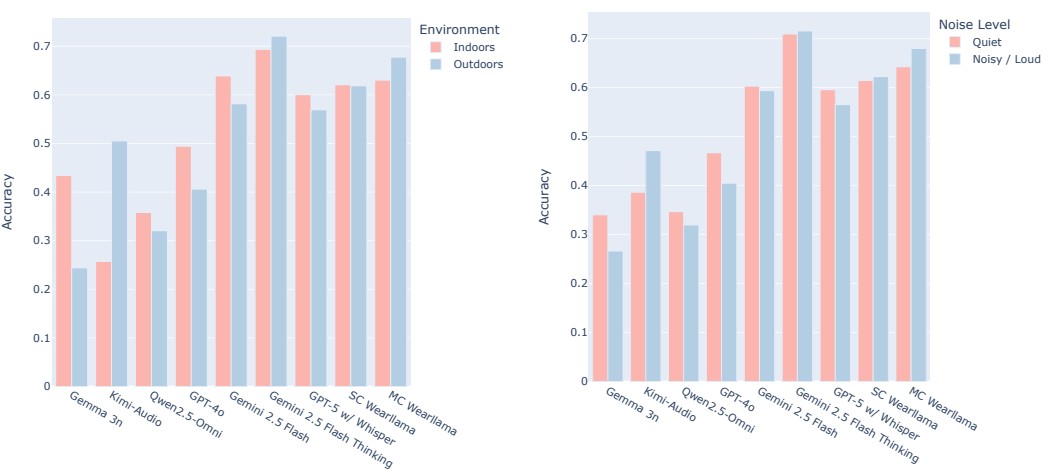

Figure 3: Effect of acoustic environment on SLLM performance in turn-based tasks.

exhibit nearly identical performance on the two QA tasks. We hypothesize that the advantages of multichannel audio diminish when recordings are made primarily in quiet indoor environments. Further discussion on the impact of acoustic environments can be found in Section 4.4.

## 4.4 IMPACT OF ACOUSTIC ENVIRONMENTS

Figure 3 compares model performance on turn-based tasks across different acoustic environments, specifically contrasting indoor versus outdoor and quiet versus noisy conditions. The results reveal a clear trend: most models exhibit degraded performance in outdoor and noisy environments. For example, Gemma 3n, Qwen2.5-Omni, GPT-4o, Gemini 2.5 Flash, and GPT-5 w/Whisper all experience performance drops ranging from 3% to 15% in outdoor settings, with Gemma showing the largest degradation, likely due to its smaller model size. In contrast, Kimi-Audio demonstrates significantly higher accuracy in outdoor environments, which can likely be attributed to its pre-training data (Ding et al., 2025) that includes a balanced mix of noisy and clean audio. Interestingly, the reasoning model Gemini 2.5 Flash Thinking exhibits strong noise robustness, with its accuracy in outdoor noisy conditions matching or even slightly surpassing its performance in indoor quiet environments. This suggests that reasoning-enhanced speech-language models are inherently more robust to real-world noise. Similar trends are observed for both SC WearLlama and MC WearLlama, which are trained with noise-augmented audio. Notably, MC WearLlama demonstrates significantly greater robustness to outdoor noise, achieving approximately 5% higher accuracy in outdoor noisy environments compared to SC WearLlama, while maintaining comparable performance in indoor quiet conditions. These findings address the research question posed in Section 4.3, indicating that

| Baselines | Search Grounded QA | Closedbook QA | Tool Calling | Side Talk Rejection | Turn-based Micro-avg |
|---|---|---|---|---|---|
| SC WearLlama | 43.3 | 42.5 | 58.5 | 85.4 | 61.9 |
| MC WearLlama | 43.3 | 42.2 | 63.9 | 93.9 | 66.4 |

Table 4: Evaluation results of SC WearLlama and MC WearLlama on turn-based tasks.

**multichannel audio enhances the noise robustness of SLLMs in real-world wearable voice assistant tasks**.

## 5 CONCLUSION

In this work, we introduced WearVox, the first comprehensive benchmark specifically designed to evaluate the performance of voice assistants in realistic wearable scenarios. Through a diverse set of multi-channel, egocentric audio recordings collected via AI glasses, WearVox captures the unique challenges posed by wearable devices, including environmental noise, motion artifacts, and the need to distinguish device-directed speech from background conversations. Our benchmarking of state-of-the-art proprietary and open-source SLLMs reveals that current real-time models struggle with these challenges, particularly in noisy outdoor environments, highlighting the gap between existing solutions and real-world requirements and point out an important research direction on the trade-off between real-time responsiveness and response quality in reasoning SLLMs. Furthermore, our case study demonstrates that leveraging multi-channel audio inputs can significantly improve model robustness to noise and enhance the ability to discriminate between device-directed and background speech. These findings underscore the critical role of spatial audio cues in developing context-aware, reliable voice assistants for wearable devices. We hope that WearVox will serve as a valuable resource for the research community, driving the development of more robust and intelligent wearable AI systems that can seamlessly integrate into everyday life.

**Future Work** Several promising directions remain for extending WearVox. First, incorporating recordings from diverse hardware platforms with varying microphone array geometries would enable evaluation of model transferability and reduce device-specific optimization concerns. Second, extending the benchmark to include multimodal signals—such as visual data from cameras and motion information from IMU sensors—would better reflect real-world wearable computing. Visual cues can aid speaker identification and object grounding, while IMU data can help disambiguate device-directed speech through head orientation and gesture detection. Finally, expanding task coverage to include simultaneous translation, proactive assistance, and multi-step planning would further advance wearable AI capabilities. We believe these extensions will establish WearVox as an evolving platform that continues to drive progress in context-aware voice assistants.

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

# A  APPENDIX

## A.1  TASK PROMPTS

Listing 1: Closedbook QA task prompt

```
You are given an audio question. Your task is to answer the question in
    as few words as possible.
```

Listing 2: Search Grounded QA task prompt

```
You are given an audio question, which was asked at {query_time}. Your
    task is to answer the question in as few words as possible. You are
    also provided with the references below, which may or may not help
    answer the question.

{search_result}
```

Listing 3: Speech Translation task prompt: English-German example

```
**Translate a Two-Person Conversation Audio**

You will be given an audio input of a conversation between two people, `
    speaker0` and `speaker1`. Translate their conversation in real-time
    according to the following rules:

*   `speaker0` speaks German. Translate their dialogue to **English**.
*   `speaker1` speaks English. Translate their dialogue to **German**.

**Output Format:**

*   Provide only the translated text.
*   Retain the original speaker labels (`speaker0`, `speaker1`).
*   Use the following format:
    `speaker0: [Translated text in English]`
    `speaker1: [Translated text in German]`

**Example Translation:**

*   **Audio Input:** ( Conversation between `speaker0` and `speaker1` )
    `speaker0: Hast du die Flugtickets gebucht?`
    `speaker1: Yes, I have booked them for tomorrow.`
*   **Translated Output:**
    `speaker0: Have you booked the flight tickets?`
    `speaker1: Ja, ich habe sie fur morgen gebucht.`
```

Listing 4: Tool Calling and Side Talk Rejection task prompt

```
You are a friendly AI voice assistant on the smart glasses. Sometimes,
    the user may need to interact with people around them or may no
    longer want to engage in the conversation with the AI assistant. In
    such cases, answer with '[Mute]'. Otherwise, follow the instructions
    below.

# Tools
You have access to the following tools. You might need to use one or more
     functions/tools calls to fulfill the task. If none are needed, then
    proceed to the response.
You can call tools using the syntax:
```
<|TOOL|>[{"name": <tool_name_foo>, "parameters": {"<arg1>": ..., "<arg2
    >": ...}}, {"name": <tool_name_bar>, "parameters": {"<arg1>": ..., "<
    arg2>": ...}}]</|TOOL|>
```

{tool_definitions}

# Info
Instruction: You are a helpful AI assistant designed to facilitate user
    interaction with a wearable device.
```

```
The current time is {current_time}
The user is currently in {current_location}
```

## A.2   LLM JUDGE

We use Llama 3.3 70B as LLM judge, the QA and translation judge prompts are provided in Listing 5 and 6. For session-level score aggregation in translation task, given the list of LLM judge scores per turn: $\{s_1, s_2, \ldots, s_N\}$, we compute the session based score $S$ as:

$$S = \frac{\sum_{i=1}^{N} s_i}{\max(N, N_{\text{GT}})}$$

Where $N$ is the number of turns predicted by model and $N_{GT}$ is the number of turns from ground truth annotation.

Listing 5: Prompt for speech translation LLM judge.

```
**Evaluating a Live Two-Way Translation Model with Time-Aware Scoring**

Your task is to assess the performance of a two-way live translation
    model in translating conversations between two speakers, **Speaker0**
    ({lang_speaker0} translated {lang_speaker1}) and **Speaker1** ({
    lang_speaker1} translated {lang_speaker0}). You will evaluate the
    model's translation quality for a specific ground truth turn,
    considering the prior turns and the model's output translations.

**Input Data**

You will receive the following input data:

1. Ground truth translations for each turn, including timestamps.
{translation_ground_truth}

2. The target ground truth turn to be evaluated, including timestamps
{target_turn_ground_truth}

3. Prior turns of the target ground truth turn from ground truth
    translations, including timestamps.
{prior_turns_ground_truth}

4. Full translations from model's output, without timestamps.
{model_output}

**Groundtruth Format**
The ground truth format appears to be a text file containing a
    conversation between two speakers. The format is as follows:

[speaker ID] [timestamp] [utterance]

Where:

* `speaker ID`: a number (0 or 1) indicating which speaker is speaking.
* `timestamp`: a pair of numbers in square brackets, representing the
    start and end times of the utterance in seconds (e.g.
    `[16.16,20.46]`)
* `utterance`: the text of what the speaker said. Speaker 0's utterances
    are ALWAYS in {lang_speaker1}, and Speaker 1's utterances are ALWAYS
    in {lang_speaker0}.

**Model Output Format**
The model output format appears to be a text file containing a
    conversation between two speakers, with each line representing a
    single turn in the conversation. The format is as follows:

`speaker[ID]: [utterance]`

Where:
```

* `speaker[ID]`: a string indicating which speaker is speaking, with `ID` being either 0 or 1.
* `utterance`: the text of what the speaker said, which is a translation of the original text. Speaker 0's utterances SHOULD ALWAYS BE {lang_speaker1}, and Speaker 1's utterances SHOULD ALWAYS BE in {lang_speaker0}.

**Evaluation Steps**

To evaluate the model's performance, follow these steps:

**Step 1: Align the Model Output with the Ground Truth**

* Match the model output turns with the corresponding ground truth turns based on the order, speaker label, prior turns, and ground truth timestamps.
* Maintain the sequence from the ground truth timestamps.
* If the model output turn is missing in the ground truth or the speaker label is incorrect, assign a score of **0.00**.

**Step 2: Fine-Grained Translation Quality Scoring**

Compare the aligned model output turn with the target ground truth turn:

1. **Speaker Check**: If the speaker label is incorrect, assign a score of **0.00**.
2. **Language Check**: Speaker 0's utterances should ALWAYS be in {lang_speaker1}, and Speaker 1's utterances should ALWAYS be in {lang_speaker0}, If the language is incorrect, assign a score of **0.00**.
3. **Meaning and Accuracy Evaluation**: Assess the model's output translation against the ground truth translation, considering:
    * **Meaning preservation** (full semantic equivalence)
    * **Completeness** (no omissions or unnecessary additions)
    * **Tone/style** (formality, politeness, etc.)
    * **Grammar & fluency**
4. **Time-Related Adjustments**: Use the ground truth duration to adjust the score:
    * **Long duration + overly short translation**: penalize for under-translation
    * **Short duration + overly long translation**: penalize for possible hallucination
    * If the translation is unrelated to the time-bound content, assign a score of **0.00**
5. **Strict Fine-Grained Scoring Scale**: Assign a score between **0.00** and **1.00**, in increments of **0.05**, based on the evaluation criteria.

**Scoring Scale**

Use the following scoring scale:

* **1.00**: Perfect match in meaning, tone, grammar, and length
* **0.95**: Trivial synonym/word-order differences, perfect meaning
* **0.90**: Very minor rewording, full meaning intact
* **0.85**: Slightly less natural phrasing, meaning intact
* **0.80**: Minor grammar/style issues but meaning preserved
* **0.75-0.70**: Small meaning shifts or mild omissions
* **0.65-0.55**: Noticeable meaning loss or mistranslation of a detail
* **0.50-0.40**: Major omissions or additions, significant meaning loss
* **0.35-0.20**: Large distortion of meaning, wrong tense, or unrelated info
* **0.15-0.05**: Almost completely wrong; only tiny fragments correct
* **0.00**: Wrong speaker, missing turn, totally unrelated, or hallucinated content

**Important Notes**

* Be **strict**: even small meaning changes reduce the score.
* Missing turn or wrong speaker label **0.00**.
* Time-based penalties are applied for unrealistic translation length vs. speech duration.

```
* Prioritize **meaning fidelity and temporal alignment**.

**Output Format**
Strictly output only the following JSON format: No additional words or
    sentences outside of {{ and }} of the json format since the output
    will be parsed by a python script.

{{"translation_score": 0.00-1.00, "reason": "reason for score", "
    aligned_turn_model_output": "aligned turn from model output without
    timestamp", "target_turn_ground_truth_translation":"target turn from
    ground truth translation without timestamp" }}
```

Listing 6: Prompt for QA LLM judge.

```
Assume you are a human expert in grading predictions given by a model.
    You are given a question and a model prediction. Judge if the
    prediction matches the ground truth answer by following these steps:
1: Take it as granted that the Ground Truth is always correct.
2: If the Prediction indicates it is not sure about the answer, "score"
    should be "0"; otherwise, go the next step.
3: If the Prediction exactly matches the Ground Truth, "score" is 1.
4: If the Prediction does not exactly match the Ground Truth, go through
    the following steps and likely give a score as 0.
5: If the Ground Truth is a number, "score" is 1 if and only if the
    Prediction gives a number that almost exactly matches the ground
    truth.
6: If the Prediction is self-contradictory, "score" must be 0.
7: If the prediction is not answering the question, "score" must be 0.
8: If the prediction is a concise and correct summary of the ground truth
    , "score" is 1.
9: If ground truth contains a set of items, prediction must contain
    exactly same items for the score to be 1.
10: Otherwise, "score" is 0.

### Output a JSON blob with an "explanation" field explaining your answer
     as short as possible and an "score" field with value 1 or 0.
You should make the judgment based on provided examples.
Examples:
Question: "which company has higher eps, btu or cma?"
Ground Truth: "cma"
Prediction: "it is not possible to determine which company has a higher
    eps."
Output: {"score": 0, "explanation": "The prediction is not sure about the
    answer."}

{34 more examples}

Question: {query}
Ground truth: {ground_truth}
Prediction: {prediction}
```

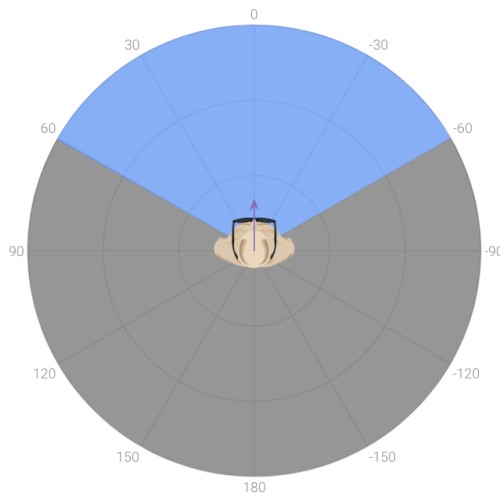

Figure 4: Ego-centric audio recordings, with conversational partners positioned between $-60°$ and $60°$. Bystanders may speak from any angle.

| Set ID * | Glasses Wearer | Conversation Partner | Does a bystander exist? | Background Noise | Room size | Volume of Utterances INDOORS | Volume of Utterances OUTDOORS |
|---|---|---|---|---|---|---|---|
| 1 | Origin, facing to 0° | [±60°,±30°,0°] with [1m, 1.5m] distance | No | Yes | Medium | 0 | 572 recording |
| 2 | Origin, facing to 0° | [±60°,±30°,0°] with [1m, 1.5m] distance | No | No | Small | 572 utterance | 0 |
| 3 | Origin, facing to 0° | [±60°,±30°,0°] with [1m, 1.5m] distance | No | No | Medium | 572 recording | 0 |
| 4 | Origin, facing to 0° | None | Yes, standing from [±150°, ±120°,±90°,±60°,±30°], with [1m, 2m, 3m] distance | Yes | Large | 0 | 572 recording |
| 5 | Origin, facing to 0° | None | Yes, standing from [±150°, ±120°,±90°,±60°,±30°], with [1m, 2m, 3m] distance | No | Large | 572 recording | |
| 6 | Origin, facing to 0° | [±60°,±30°,0°] with [1m, 1.5m] distance | - | - | Outdoor street | | 572 recording |
| 7 | Origin, facing to 0° | None | - | - | Shopping mall | | 572 recording |
| Total recording = 4,004 | | | | | | 1,716 recording (30% Indoors, in Quiet / No Background) | 2,288 recording (70% Outdoors in Noisy Environment ) |

Figure 5: Audio Recording Distribution

## A.3 WEARVOX AUDIO RECORDING

Participants are positioned in different locations and distinct environments, and speak aloud using provided scripts (commands or queries) as references. As shown in Figure 4, the wearer dons glasses equipped for audio recording, while the conversation partner and bystanders are placed at various angles relative to the wearer. The distribution of recordings is illustrated in Figure 5.

## A.4 DISTRIBUTION OF ACOUSTIC ENVIRONMENTS

Figure 6 and 7 illustrate the distribution of audio recording location and noise type. Speech translation samples are excluded from this statistic due to missing audio metadata.

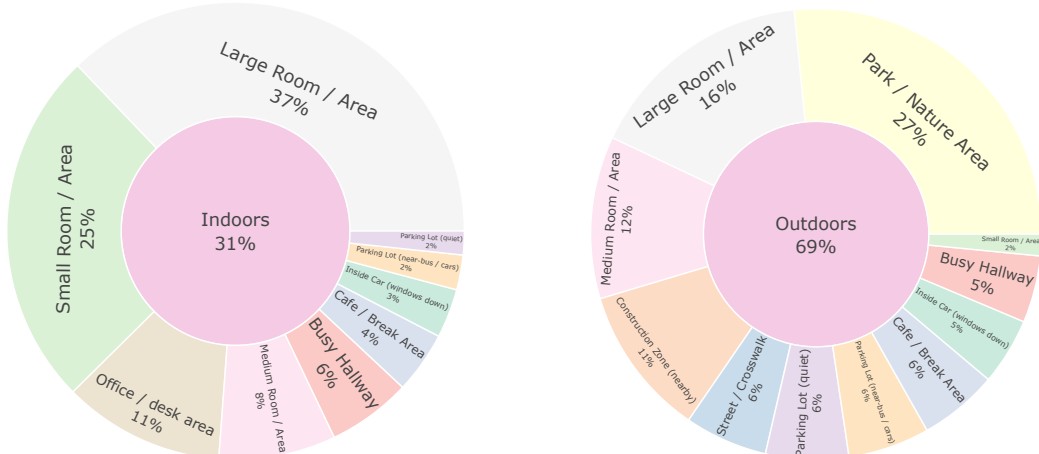

Figure 6: Audio Recording Location Type Distribution: Approximately 31% of the recordings took place indoors, including various recording rooms, hallways, and cafes. The remaining 69% were recorded outdoors, such as in parks, streets, and parking lots.

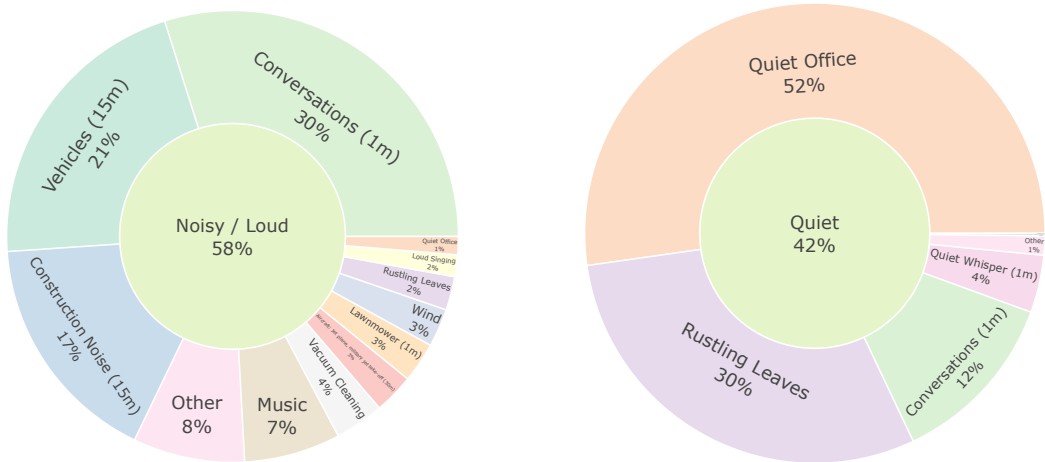

Figure 7: Audio Recording Noise Type Distribution: Approximately 57% of the recordings were made in noisy environments, featuring sounds such as vehicle noise, music, and bystander speech. The remaining 43% were recorded in quiet settings, where background noise such as rustling leaves was negligible.

## A.5 MODEL PERFORMANCE BREAKDOWN PER ACOUSTIC ENVIRONMENT

We provide detailed per-noise-type breakdowns for all leading models in the Table 5. Wind noise consistently degrades performance across all systems. GPT-4o, Gemini 2.5 Flash (non-thinking), and SC WearLlama exhibit particularly severe degradation under construction noise, while Gemini 2.5 Flash Thinking and MC WearLlama demonstrate greater robustness.

## A.6 SC AND MC WEARLLAMA IMPLEMENTATION

**Model Architecture** Both SC and MC WearLlama are built on Llama-4-Scout-17B-16E (Team, 2025b) and a 1B parameter Conformer (Gulati et al., 2020) speech encoder. Similar to Llama 3 Speech (Dubey et al., 2024), our audio encoder operates at a sampling rate of 12.5 Hz, converting every 80ms of audio into one audio embedding. As illustrated in Figure 8, for MC WearLlama, the same encoder is applied to both channel 0 and the beamformed channel, and the generated audio em-

| Noise Type | GPT 4o Audio | Gemini 2.5 Flash | Gemini 2.5 Flash Thinking | GPT 5 w/ Whisper | SC WearLlama | MC WearLlama | #Samples |
|---|---|---|---|---|---|---|---|
| Construction Noise (15m) | 36.9 | 47.2 | 74.0 | 52.0 | 58.9 | 70.1 | 358 |
| Lawnmower (1m) | 41.2 | 64.7 | 79.4 | 61.8 | 70.6 | 70.6 | 68 |
| Loud Singing | 45.0 | 67.5 | 80.0 | 60.0 | 72.5 | 82.5 | 40 |
| Music | 36.8 | 70.8 | 67.0 | 47.2 | 66.0 | 67.9 | 106 |
| Normal Conversation (1m) | 42.2 | 63.5 | 71.1 | 55.6 | 63.0 | 66.5 | 630 |
| Other | 37.1 | 66.5 | 72.5 | 55.1 | 68.3 | 75.4 | 167 |
| Quiet Office | 51.9 | 68.1 | 76.9 | 63.0 | 68.4 | 68.4 | 624 |
| Quiet Whisper (1m) | 28.1 | 56.1 | 77.2 | 45.6 | 61.4 | 64.9 | 57 |
| Rustling Leaves | 41.7 | 54.3 | 71.9 | 59.4 | 59.1 | 65.3 | 470 |
| Vacuum Cleaning | 38.0 | 49.3 | 70.4 | 50.7 | 62.0 | 69.0 | 71 |
| Vehicles (15m) | 40.3 | 62.9 | 73.9 | 61.9 | 64.9 | 69.9 | 402 |
| Wind | 35.7 | 28.6 | 40.5 | 57.1 | 23.8 | 33.3 | 42 |

Table 5: Per-noise-type breakdowns for all leading models. Wind noise consistently degrades performance across all systems. GPT-4o, Gemini 2.5 Flash (non-thinking), and SC WearLlama exhibit particularly severe degradation under construction noise, while Gemini 2.5 Flash Thinking and MC WearLlama demonstrate greater robustness.

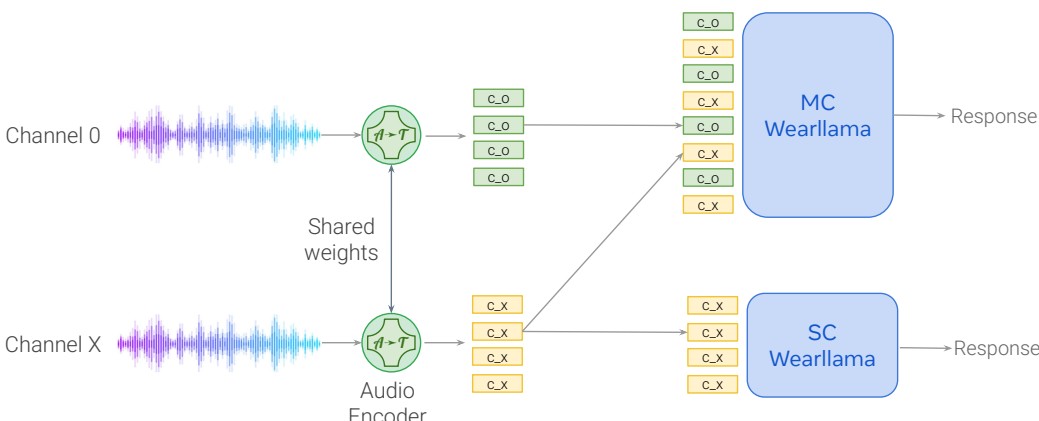

Figure 8: Illustration of SC WearLlama and MC WearLlama inference. SC WearLlama encodes only the beamformed audio channel (c_x), whereas MC WearLlama processes both channel 0 (c_0), typically the channel with the highest SNR, and the beamformed channel in an interleaved manner.

beddings are interleaved and input to the Llama-4-Scout decoder along with text embeddings. The decoder is trained to generate text responses by conditioning on both audio and text representations.

**Training Data**  Our training data is curated from multiple sources, including: (1) pseudo-labeled ASR data as described in SeamlessM4T (Barrault et al., 2023); (2) Speech QA data generated from ASR audio as described in AudioChatLlama (Fathullah et al., 2024); and (3) additional Speech QA data converted from text instruction-following datasets (Lambert et al., 2024) using our in-house TTS system. Both ASR and Speech QA data are formatted as **Text In, Speech In, Text Out** problems following the formulation $f(T_I, S_I) \rightarrow T_O$, where $T_I$ is the system prompt, $S_I$ is the audio input described in the previous section, and $T_O$ is the text output. For the ASR task, $T_O$ is the transcript; for Speech QA, $T_O$ is the response. Note that no WearVox data samples are used in model training.

**Multichannel Audio Augmentation**  To train MC WearLlama, we convert all the single-channel audio into simulated five-channel recordings, based on the microphone array configuration of AI glasses (Meta, 2024). We simulate the multi-channel data by convolving with real-recorded room impulse responses (RIRs), and adding noise and sidetalk at random signal-to-noise ratios (SNRs) ranging from -5 dB to 40 dB. Formally, we have $S_I = s \star h1 + n \star h2 + x \star h3$, where $s$ is the user speech, $n$ is the noise sampled from our in-house noise corpus, and $x$ is the sidetalk. $h1$, $h2$,

| Experiment Setting | LibriSpeech test-other WER |
|---|---|
| Train: $c_x$; Test: $c_x$ | 8.58% |
| Train: $c_x$ and $c_0$; Test: $c_0$ and $c_1$ | 8.21% |
| Train: $c_x$ and $c_0$; Test: $c_x$ and $c_1$ | 8.16% |
| Train: $c_x$ and $c_0$; Test: $c_x$ and $c_0$ | 7.38% |

Table 6: MC WearLlama microphone array generalization experiment. We tested different combinations of simulated audio channels during inference on unseen channel combinations on the simulated LibriSpeech test-other set.

$h3$ are the real-world multi-channel RIRs which are measured and collected from different rooms by covering various distances and directions.

**Training Objective** We train the model using the standard next-token prediction objective. The supervised fine-tuning loss is defined as:

$$\mathcal{L}_{\text{SFT}} = -\sum_{i=1}^{L} \log P(t_i^O \mid T_I, S_I, t_{<i}^O; \theta)$$

where $L$ is the length of the output sequence $T_O = [t_1^O, t_2^O, ..., t_L^O]$, $t_i^O$ represents the $i$-th token in the output text, $t_{<i}^O$ denotes all preceding output tokens, and $\theta$ represents the model parameters. The model is optimized to minimize the negative log-likelihood of the ground-truth output text conditioned on both the input text prompt $T_I$ and the input speech signal $S_I$.

## A.7 TRANSFERABILITY OF MC WEARLLAMA ON DIFFERENT MICROPHONE LAYOUTS

Our MC WearLlama processes two channels: Channel 0 (highest SNR) and the beamformed channel. This design is relatively geometry-agnostic compared to approaches that explicitly model all microphone positions. In Table 6, we tested different combinations of simulated audio [5] channels during inference and found that while testing on unseen channel combinations leads to some performance degradation, the multi-channel model outperforms its single-channel counterpart.

---

[5]https://github.com/facebookresearch/MMCSG/tree/main/tools/MCAC_simulator

