# OpenReview forum: "WearVox: An Egocentric Multichannel Voice Assistant Benchmark for Wearables"
_ICLR.cc/2026/Conference — ICLR 2026 Poster_

### Official Review · Reviewer_6q8y · 2025-10-31

**Soundness:** 3
**Presentation:** 3
**Contribution:** 3
**Rating:** 8
**Confidence:** 4

**Summary:**

This paper collects a large set of in-the-wild data of users interacting with expected Voice AI tasks using microphone enabled glasses. The range of tasks tested is Search-Grounded QA, Closed-Book QA, Side-Talk Rejection, Tool Calling, and Speech Translation. For each task, there are multiple possible speakers including the wearer, the conversational partner, and bystanders and speech from each should be handled differently on a contextual basis. The authors first benchmark a range of current audio models on these benchmarks, then do an ablation of their own custom models designed for this data type and find that multi-channel signal processing is key to handling noise and acoustic cues in the data.

**Strengths:**

- The dataset covers realistic tasks, realistic environments, and realistic challenges for speech models. Despite being primarily focused on this realism, the benchmark is clearly difficult for frontier speech LLMs especially open-weights speech LLMs. It will clearly help measure (and therefore drive) progress.
- The multi-party nature of the environment is a welcome addition to the speech benchmarking space which usually operates on data from relatively clean single-speaker environments. With this, the benchmark will clearly drive models to be more robust to the natural challenges from real multi-party environments (such as distractions and interruptions) that often plague today's speech LLMs.
- The authors then go above and beyond reporting the results of their benchmark by showing an example of a low-level ablation which the benchmark allows them to enable the impact of.

**Weaknesses:**

- The benchmark seems somewhat likely to incentivize models to optimize for the specific hardware the benchmark was recorded on. This is of course true of any speech benchmark which uses a consistent recording device, but might become increasingly impactful for devices which have specific form factors and layouts for their microphone arrays.

**Questions:**

- Are there any ablations for the SC and MC Llama on how much more specified MC Llama is to the specific layout of the microphones in these particular glasses? Could you train MC Llama on a simulations of a different Multi-Channel layout and get similarly strong transfer to WearVox? While it makes sense that multi-channel is useful, it's not clear to what degree a single channel model might be more transferrable across devices.

- Are you able to share the participant demographics mentioned in the ethics statement? It says "demographic data was aggregated and used solely for fairness analysis." but I wasn't able to find such analysis and it would be valuable to share.

(Typo on 347. Appedix should be Appendix)

---

> ### Author Response · Authors · 2025-11-21
>
> We sincerely thank the reviewer for their thorough evaluation and positive feedback. We appreciate your recognition of the realism and difficulty of WearVox, its multi-party nature as a welcome addition, and our efforts to provide comprehensive ablations. Below, we address the raised concerns:.
>
> ## 1. The benchmark may incentivize models to optimize for the specific microphone array layout of the recording device.
>
> We appreciate this important observation. We agree that hardware-specific optimization is a valid concern for any audio benchmark using consistent recording equipment. In our updated paper, we have added hardware expansion as an important direction for the future work. We also would like to highlight that the concern is partially mitigated in our case for several reasons:
> - Beamforming as standardization: For single-channel baselines, we apply beamforming (Section 4.1.1), which is a standard signal processing technique that generalizes across different microphone array geometries. This allows single-channel models to benefit from multi-mic recordings without being tied to specific hardware.
> - Multi-channel WearLlama: Our MC WearLlama is trained on simulated multi-channel data with the beamformed channel x and the channel with highest SNR (channel 0). This training approach, rather than being hardware-specific, teaches the model to leverage spatial audio cues more generally.
> - Evaluation focuses on capabilities: WearVox primarily evaluates models' ability to handle realistic acoustic challenges (noise, side-talk, multi-speaker scenarios) rather than hardware-specific signal processing. The benchmark's value lies in measuring these context-aware capabilities.
>
> ## 2. Transferability of MC Llama across different microphone layouts
> This is an excellent research question. While we don't have direct ablations on WearVox., we can provide the following insights:
> - Our MC Wearllama processes two channels: Channel 0 (highest SNR) and the beamformed channel (Section 4.3, Figure 2). This design is relatively geometry-agnostic compared to approaches that explicitly model all microphone positions.
> - The model is trained on simulated multi-channel data with diverse RIRs, which introduces variability in spatial characteristics and should promote some degree of layout generalization.
> - Prior work on array-geometry agnostic models (Lin et al., 2024, cited in our paper) suggests that learning spatial features can transfer across different array configurations when properly designed.
> - We tested different combinations of simulated audio channels during inference and found that while testing on unseen channel combinations leads to some performance degradation, the multi-channel model still outperforms its single-channel counterpart.
> | Experiment Setting| LibriSpeech test-other WER|
> |:-----------|--------------------:|
> | Train: C_x Test: C_x| 8.58%|
> | Train: C_x & C_0 Test: C_0 & C_1| 8.21%|
> | Train: C_x & C_0 Test: C_x & C_1| 8.16%|
> | Train: C_x & C_0 Test: C_x & C_0| 7.38%|

---

> > ### Comment · Reviewer_6q8y · 2025-11-21
> >
> > Thanks for a set of measured and helpful responses! While it doesn't lead me to increase my score, it does increase my confidence that this is a valuable dataset and evaluation.
> >
> > The ablations on other simulated audio channels is helpful at estimating the degree of effect specific layouts should be expected to have and will be useful if included in the appendix. It would further be useful if you release the software used to simulate these conditions so others can perform similar analyses.
> >
> > Overall, I am quite strongly of the opinion, in contrast to some other reviewers, that even the single channel version of this benchmark would be valuable to speech language model developers in academia and industry in that it captures many realistic phenomena that existing benchmarks lack! The wearable aspect of the dataset adds some further utility.

---

> ### Author Response · Authors · 2025-11-21
>
> We greatly appreciate your confidence in the value of our dataset and evaluation.
>
> We have included the ablations on simulated audio channels in the appendix A.7 as you suggested, and have linked to an open-sourced multichannel simulation software to enable other researchers to perform similar analyses and extend this work to their own contexts.

---

### Official Review · Reviewer_tmBp · 2025-10-31

[review text omitted: it was posted to a different submission]

---

> ### Author Response · Authors · 2025-11-21
>
> We sincerely thank the reviewer for their time and detailed feedback. However, we noticed some significant discrepancies between the review and our submitted paper that we would like to clarify before addressing the applicable questions.
> # Clarification on Paper Content
> We respectfully note that several aspects of the review appear to describe a different work than our submission:
>
> The review mentions:
> - RGB video, binaural microphones, and body-contact microphones
> - Tasks including speaker localization and audio-visual event detection
> - Focus on egocentric vision and audio-visual perception
> - A model called "WearVoxNet" with audio-visual encoders
>
> Our paper presents:
> - Multi-channel audio only (no video components) from AI glasses microphone arrays
> - Tasks: Search-Grounded QA, Closed-Book QA, Side-Talk Rejection, Tool Calling, and Speech Translation
> - Focus specifically on voice assistant evaluation for wearable devices
> - A case study with SC/MC Wearllama (Section 4.3) to demonstrate multi-channel benefits
>
> Thus there are several concerns that are not applicable to our work (e.g., Weakness 1 on "WearVoxNet model novelty," Weakness 3 on "contact microphone analysis," and Weakness 6 on "modality contribution across tasks").
>
> We suspect there may have been a mix-up in the review assignment. Nevertheless, we address the questions and concerns that are applicable to our actual submission below.
> # Responses to Applicable Questions and Concerns
> ## Question 1: Cross-Dataset Generalization Experiments
> While we appreciate this suggestion, we'd like to clarify our paper's scope and contribution:
> **WearVox is primarily a benchmark dataset, not a new model or new training data**. Our main goal is to establish a rigorous evaluation testbed for voice assistants in realistic wearable scenarios - a gap we identified in existing benchmarks (Table 1).
> Regarding cross-dataset evaluation:
> - Ego4D and AVD focus on different modalities (vision-centric tasks) and are not directly comparable to our audio-only voice assistant tasks
> - Our contribution is evaluating how well existing, pre-trained SLLMs perform on wearable scenarios out-of-the-box, which is orthogonal to transfer learning experiments
> That said, our MC Wearllama case study (Section 4.3) does demonstrate a form of domain transfer: we train on simulated multi-channel audio with augmented noise and RIRs, then evaluate on real-world WearVox recordings, showing that multi-channel spatial cues improve robustness.
> ## Weakness 4: Practical Deployment Constraints
> We thank the reviewer for raising this important point. Our experiments primarily focus on cloud-based inference rather than on-device deployment, as this reflects the most common architecture for current wearable voice assistants where audio is streamed from the device to cloud servers. This design choice also ensures our benchmark remains device-agnostic, given the substantial variation in compute capacity across different AI glasses hardware.
>
> Regarding the latency constraints, we have added the per-task TTFT (in ms) breakdown for Gemini 2.5 Flash with and without thinking mode, as well as GPT-4o Audio. We observe that the thinking model exhibits significantly higher latency compared to its non-thinking counterpart and suggested that non-thinking model is more suitable for wearable use cases.
>
> | Task Type| Gemini 2.5 Flash| Gemini 2.5 Flash Thinking |GPT4o Audio |
> |:-----------------------|-----------------------------------------------:|--------------------------------------------------------:|----------------------------------------:|
> | Closedbook QA|1368.69 | 2287.76 | 1220.22 |
> | Search Grounded QA |1526.56 | 9194.94 | 1867.66 |
> | Speech Translation |2138.11 |11321.49 | 7523.24 |
> | Side Talk Rejection |1306.62 | 2176.97 | 1341.04 |
> | Tool Calling |1404.69 | 2084.19 | 1289.99 |
>
> ## Weakness 5: Empirical Comparisons
> - We respectfully believe our empirical evaluation is comprehensive within the voice assistant domain:
> - We benchmark 7 state-of-the-art SLLMs: GPT-4o Audio, Gemini 2.5 Flash, GPT-5 w/ Whisper, Qwen2.5-Omni, Kimi-Audio, Gemma 3n
> - These represent the current frontier in speech-to-text-to-speech and end-to-end audio LLMs
> - We include both proprietary and open-source models for balanced comparison
>
> The suggested benchmarks (AVD, Ego4D) focus on audio-visual or vision-centric tasks that are fundamentally different from our voice assistant evaluation. To our knowledge, no prior work provides comparable baselines for wearable voice assistant tasks as defined in our benchmark.

---

### Official Review · Reviewer_Nq5Q · 2025-11-01

**Soundness:** 3
**Presentation:** 2
**Contribution:** 3
**Rating:** 4
**Confidence:** 4

**Summary:**

This paper introduces WearVox, a benchmark dataset designed to evaluate voice assistants in realistic wearable scenarios, e.g. with wearable glasses. The authors mention that existing voice assistant benchmarks fail to capture challenges of wearable, egocentric audio such as motion, wind noise, background and side-talk rejection. This paper provides a benchmark dataset of 3,842 recordings captured using AI glasses in multi-channel audio format for five diverse tasks that AI assistants may have to do: Search-Grounded QA, Closed-Book QA, Side-Talk Rejection, Tool Calling, and Speech Translation. The authors study existing voice assistant methods, Speech Large Language Models (SLLMs) as baselines and also introduce specialized new models that handle native multichannel audio better.

**Strengths:**

1. This paper is a solid contribution to benchmark real-world AI assistant applications in a wearable setting, such data is very hard to find and is expensive to manually curate, script, and collect. It is a first of a kind dataset for this emerging setting in HCI and human-AI interfaces.

1. The baselining is done for a wide range of commercial models, and is done across settings and tasks that matter for this wearable setting.

1. "Side-Talk Rejection" is arguably the single most important and overlooked challenge for an always-on wearable assistant. Evaluating this explicitly is a huge contribution.

1. The paper carefully curates data, using specific angles and distances, and capturing audio across 13 different noise types, the paper has created a dataset that reflects the structured complexity of real-world interactions, not just random noise.

**Weaknesses:**

1. For the custom trained models, it is not clear if there is data leakage? It is also not clear from the paper what data was used to train these models. Presumably the baseline models were also trained with noise augmentation as is standard with commercial-grade speech models.

1. There doesn't seem to be an easy way to explore the benchmark. Perhaps this is a way to prevent leakage, but it would be great to be able to explore this dataset in an interactive way.

1. It is hard to get a sense of diversity in the dataset. Presumably the annotators who were asked to write multi-turn conversations did a good job, but there are no quantitative evaluations about the _quality_ or diversity of this dataset. Additionally, the limited scale might not capture edge cases that matter like speaker accents.

1. The data was collected using the microphone array from a specific set of AI glasses, and does not capture the diversity in hardware.

**Questions:**

1. More details on SC/MC WearLlama would be nice, especially the training data and methods. Was there any overlap or fine-tuning done on WearVox? You demonstrated that proprietary models struggled with noise. How did your noise augmentation strategy for Wearllama differ from the standard techniques you assume are used in commercial models?

1. Is there a way to measure the "realness" and diversity of the scripts? Particularly humans are bad at creating entropy, what measures were takes to make sure sure there is enough diversity?

1. Are there plans to make the WearVox dataset, or at least a sample of it, publicly available or browseable? An interactive way to explore the audio and metadata would be invaluable for the research community.

1. You used an LLM to judge the QA and Translation tasks. What steps did you take to validate the LLM-judge itself? For instance, how did its scores correlate with human expert scores on a subset of the data?

---

> ### Author Response · Authors · 2025-11-21
>
> We sincerely thank the reviewer for the thoughtful and constructive feedback. We appreciate your recognition of WearVox as a "solid contribution" and acknowledgment of its importance for wearable AI assistants. Below, we provide a point-by-point response to each concern:
> ## 1. Data Leakage & Training Data/Methods Details on SC/MC WearLlama.
> We would like to highlight that **WearVox was held out entirely and used only for evaluation**. We have added the more model/data/training details of SC/MC WearLlama in our updated paper **Appendix A.6**. (highlighted in red)
> ## 2. Ways to Explore WearVox & Release Plan
> We’ve uploaded some audio examples for each task in the **Supplementary Material**. We will **release the whole dataset and evaluation scripts** shortly after the paper decision date.
> ## 3. Diversity & Quality Metrics
> Good point. Here are the quantitative diversity metrics:
>
> **Linguistic Diversity (Scripts diversity):**
>
> We acknowledge humans are indeed "bad at creating entropy"—this is why we combined human creativity with LLM query expansion during script collection as discussed in paper Section 3.2.
> - Lexical diversity:  To demonstrate the “realness” and diversity of the scripts. We random sample 4000 audio samples from popular audio benchmark Multilingual LibriSpeech (MLS) eval set and compare the unique n-grams in the transcripts. As shown in the table, Wearvox show higher lexical diversity.
> | Dataset | #Samples | Unique Unigrams | Unique Bigrams | Unique Trigrams |
> |-------------------------|----------|-----------------|----------------|-----------------|
> | Wearvox | 4000 | 27135 | 122914 | 180862|
> | Multilingual LibriSpeech| 4000 | 14595 | 83773| 124525|
>
> - Domain Diversity: As discussed in the main paper, Wearvox also features diverse domains: 14 domains/topics:
> | domain | sample_percentage |
> |:-----------|------------:|
> | Local| 0.274857|
> | Other| 0.216033|
> | Music| 0.0950026 |
> | Finance| 0.0921395 |
> | Movie| 0.0895367 |
> | Sports | 0.0845914 |
> | Reminder | 0.0515357 |
> | Calendar | 0.0340968 |
> | Weather| 0.0275898 |
> | People | 0.0119729 |
> | Books| 0.00780843|
> | News | 0.00754815|
> | Navigation | 0.00702759|
> | Time | 0.000260281 |
>
> **Acoustic Diversity:**
>
> - 13 distinct noise types at SNRs from -5dB to 40dB (documented)
> - Speaker variation: 100 participants across 5 language pairs
> - Microphone array positions: 6 geometric configurations (Appendix A.3)
> - Regarding accents: We acknowledge this as a limitation. Our participants were primarily from North America and Europe. We will note this in limitations and propose accent diversity as important future work.
>
> **Quality Control:**
>
> We performed both automated and manual quality checks to ensure data quality. Our process included the following steps:
> Automated Checks:
> - LLM-Based Query Review: We leveraged LLMs to flag samples requiring rewriting before conversion to spoken form (e.g., those containing dates or brackets). These samples were then revised with LLM assistance.
> - Manual Quality Assurance: We then manually reviewed the data and removed approximately 10% of samples that were unclear, incorrectly labeled, or outside the scope of our predefined tasks.
> ## 4.The data was collected using the microphone array from a specific set of AI glasses, and does not capture the diversity in hardware
> We agree that hardware-specific optimization is a valid concern for any audio benchmark using consistent recording equipment. In our updated paper, we have added hardware expansion as an important direction for the future work. We also would like to highlight that the concern is partially mitigated in our case for several reasons:
> - Beamforming as standardization: For single-channel baselines, we apply beamforming (Section 4.1.1), which is a standard signal processing technique that generalizes across different microphone array geometries. This allows single-channel models to benefit from multi-mic recordings without being tied to specific hardware.
> - MC WearLlama microphone array generalization: We tested different combinations of simulated audio channels during inference and found that while testing on unseen channel combinations leads to some performance degradation, the multi-channel model still outperforms its single-channel counterpart.
> | Experiment Setting| LibriSpeech test-other WER|
> |:-----------|--------------------:|
> | Train: C_x Test: C_x| 8.58%|
> | Train: C_x & C_0 Test: C_0 & C_1| 8.21%|
> | Train: C_x & C_0 Test: C_x & C_1| 8.16%|
> | Train: C_x & C_0 Test: C_x & C_0| 7.38%|
>
> ## 5. LLM-judge Validation
> We have added the LLM judge validation section in the paper (4.1.2): The LLM-based judge for QA tasks was adapted from the auto-evaluation of CRAG, which has demonstrated over 98% agreement with human evaluation. For the translation task, we validated the judge on 200 randomly sampled examples and observed a strong Pearson correlation (r = 0.89) between our judge's scores and human ratings, with human raters using the same scoring scale as the judge.

---

### Official Review · Reviewer_muyQ · 2025-11-03

**Soundness:** 3
**Presentation:** 3
**Contribution:** 3
**Rating:** 6
**Confidence:** 4

**Summary:**

WearVox introduces a wearable-specific benchmark of 3842 egocentric multichannel recordings collected with AI glasses, spanning 5 speech tasks, with each clip having rich environment and position metadata for nuanced analysis. The paper benchmarks open-source and proprietary SLLMs and reports large drops in noisy/outdoor conditions.

**Strengths:**

- First benchmark aimed squarely at wearables with egocentric, multi-mic audio, diverse indoor/outdoor scenes, and explicit side-talk; prior suites largely miss these factors.
- Both open and proprietary SLLMs; headline finding: most real-time SLLMs land ~29–59% on WearVox, highlighting difficulty.
- Five tasks with clean input/output definitions.
- Support multi-channel processing, for testing the SLLMs.

**Weaknesses:**

- Some reporting is aggregate. More per-environment/per-distance breakdowns (beyond the figures) would make failure modes easier to act on.
- "Thinking" boosts scores but increases TTFT substantially, this deserves heavier emphasis for wearables.
- A careful proofreading pass is needed to improve clarity as well as typos.
- No examples to listen.

**Questions:**

- Will you release a public leaderboard and evaluation server? If so, please provide the planned URL and timeline.
- Please report per-task latency and time-to-first-token (TTFT) for each model—especially in ‘thinking’ mode—so we can judge real-time feasibility on wearables.
- Could you please eleberate more on the MC Wearllama? The description is not clear.

---

> ### Author Response · Authors · 2025-11-21
>
> We sincerely thank the reviewer for their thoughtful evaluation. We appreciate the recognition of WearVox as the "first benchmark aimed squarely at wearables" with valuable contributions. Below, we provide a point-by-point response to each concern:
> ## 1. More per-environment/per-distance breakdowns
> We have added the following in our paper **Appendix A5**:
> | Noise Type |GPT4o Audio |Gemini 2.5 Flash | Gemini 2.5 Flash Thinking |GPT 5 w/ Whisper |SC WearLlama |MC WearLlama |#Samples |
> |:--------------------------------------------|-----------------------:|-------------------------------:|----------------------------:|------------------:|--------------------:|--------------------:|--------:|
> | Construction Noise (15m) | 36.9 |  47.2 |  74 |  52 | 58.9 |  70.1 |   358 |
> | Lawnmower (1m) | 41.2 |  64.7 |  79.4 |  61.8 |  70.6 |  70.6 | 68 |
> | Loud Singing  | 45 | 67.5 | 80 | 60 | 72.5 |82.5 |40 |
> | Music| 36.8 | 70.8 |67 |47.2 |  66 |  67.9 | 106 |
> | Normal Conversation (1m) | 42.2 | 63.5 | 71.1 | 55.6 | 63| 66.5 | 630 |
> | Other  |  37.1 | 66.5 |72.5 |55.1 |68.3 |75.4 | 167 |
> | Quiet Office| 51.9 | 68.1 |76.9 |63 |68.4 |68.4 | 624 |
> | Quiet Whisper (1m)| 28.1 | 56.1 |77.2 |45.6 |61.4 |64.9 |57 |
> | Rustling Leaves | 41.7 | 54.3 |71.9 |59.4 |59.1 |65.3 | 470 |
> | Vacuum Cleaning | 38 | 49.3 |70.4 |50.7 |62 |69 |71 |
> | Vehicles (15m) | 40.3 | 62.9 |73.9 |61.9 |64.9 |69.9 | 402 |
> | Wind| 35.7 | 28.6 |40.5 |57.1 |23.8 |33.3 |42 |
>
> We provide detailed per-noise-type breakdowns for all leading models in the Table. Wind noise consistently degrades performance across all systems. GPT-4o, Gemini 2.5 Flash (non-thinking), and SC Wearllama exhibit particularly severe degradation under construction noise, while Gemini 2.5 Flash Thinking and MC Wearllama demonstrate greater robustness.
> ## 2. Latency analysis for thinking vs non-thinking models
> Thanks for the suggestion, we have added the per task per-task TTFT (in ms) breakdown in our *paper (section 4.2)*. Below we show the TTFT for Gemini 2.5 Flash with and without thinking mode, as well as GPT-4o Audio. We observe that the thinking model exhibits significantly higher latency compared to its non-thinking counterpart, primarily due to the overhead of thinking token generation. GPT-4o Audio demonstrates comparable latency to Gemini 2.5 Flash across most tasks, with the notable exception of speech translation, where the slower audio encoding during prefill likely contributes to increased delay.
> | Task Type| Gemini 2.5 Flash| Gemini 2.5 Flash Thinking |GPT4o Audio |
> |:-----------------------|-----------------------------------------------:|--------------------------------------------------------:|----------------------------------------:|
> | Closedbook QA|1368.69 | 2287.76 | 1220.22 |
> | Search Grounded QA |1526.56 | 9194.94 | 1867.66 |
> | Speech Translation |2138.11 |11321.49 | 7523.24 |
> | Side Talk Rejection |1306.62 | 2176.97 | 1341.04 |
> | Tool Calling |1404.69 | 2084.19 | 1289.99 |
> ## 3. Audio Samples and Dataset Release
> We’ve uploaded some audio examples for each task in the **Supplementary Material**. We will **release the whole dataset and evaluation scripts** shortly after the paper decision date.
> ## 4. Proofreading & clarity improvement & typos fixes
> We have
> - 1) fixed identified typos (e.g., "snarios" → "scenarios" in line 107)
> - 2) clarified the MC Wearllama technical descriptions in our updated paper **Appendix A.6**

---

### Meta-Review · Area_Chair_2FHg · 2026-01-07

**Summary:**

The decision process for this paper is looking very positive, though there was one massive confusion to clear up first. Reviewer tmBp appears to have reviewed a completely different submission, criticizing the lack of analysis on "contact microphones" and "video" features that simply do not exist in this audio-only paper. For the valid reviews, the main concerns focused on whether the benchmark was too tailored to specific hardware and if the "thinking" models were too slow for real-time wearables. The authors handled these well by providing latency breakdowns and proving that their multi-channel approach generalizes beyond the specific glasses used for recording

**Reviewer Concerns:**

Addressed by Rebuttal:
The "Wrong Paper" Mix-up (Reviewer tmBp): This reviewer hallucinated features like "body-contact microphones" and "RGB video." The authors clarified that the paper is audio-only, which invalidates almost all of this reviewer's critiques regarding novelty and modality analysis.

Hardware Overfitting (Reviewer 6q8y & Nq5Q): There was a fear that the results only applied to the specific mic array on the glasses used. The authors mitigated this by showing how beamforming standardizes the input and provided simulated ablations to prove the models aren't just memorizing specific mic geometries.
​
Latency in "Thinking" Models (Reviewer muyQ): The reviewer needed to know if "thinking" models are viable for wearables. The authors provided a clear table showing a huge latency cost (e.g., jumping from ~2s to ~11s for translation), which effectively answered the question.

Outstanding:

Demographic Diversity (Reviewer Nq5Q): The reviewer noted a lack of accent diversity. The authors admitted their participants were mostly North American/European and just listed this as a limitation for future work rather than fixing it

Cross-Dataset Generalization (Reviewer tmBp): The reviewer asked for tests on video datasets like Ego4D. The authors argued this was out of scope for an audio benchmark, so no new experiments were run here.

**Reviewer Scores:**

Reviewer 6q8y: They explicitly stated that while the rebuttal was helpful and increased their confidence in the paper's value, it wasn't enough to push their score higher than the "Accept" they already gave

Reviewer muyQ:  The authors provided the exact data breakdowns and latency analysis requested. Since the reviewer was already leaning positive, these additions should be enough to bump them up to a solid accept

Reviewer Nq5Q: This reviewer was on the fence mostly due to data leakage fears and hardware specificity. The authors clarified that the test set was held out (no leakage) and addressed the hardware concerns, which should be enough to flip them to a weak accept

Reviewer tmBp: It's hard to predict because they reviewed the wrong paper. If they actually read the correct paper now, they might stick to a 6 since the actual paper is still a solid benchmark, even if it lacks the "contact mics" they imagined

---

### Decision · Program_Chairs · 2026-01-26

Accept (Poster)